# Postnatal corticosteroid use for prevention or treatment of bronchopulmonary dysplasia in England and Wales 2012–2019: a retrospective population cohort study

Sijia Yao,[1] Sabita Uthaya ![ORCID],[2] Chris Gale ![ORCID],[2] Neena Modi ![ORCID],[2] Cheryl Battersby ![ORCID],[2] On behalf of the UK Neonatal Collaborative (UKNC)

[1]Neonatal Medicine, Imperial College London, London, UK
[2]Neonatal Medicine, School of Public Health, Imperial College London, London, UK

**Correspondence to**
Dr Cheryl Battersby;
c.battersby@imperial.ac.uk

## ABSTRACT

**Objective** Describe the population of babies who do and do not receive postnatal corticosteroids for prevention or treatment of bronchopulmonary dysplasia (BPD).

**Design** Retrospective cohort study using data held in the National Neonatal Research Database.

**Setting** National Health Service neonatal units in England and Wales.

**Patients** Babies born less than 32 weeks gestation and admitted to neonatal units from 1 January 2012 to 31 December 2019.

**Main outcomes** Proportion of babies given postnatal corticosteroid; type of corticosteroid; age at initiation and duration, trends over time.

**Secondary outcomes** Survival to discharge, treatment for retinopathy of prematurity, BPD, brain injury, severe necrotising enterocolitis, gastrointestinal perforation.

**Results** 8% (4713/62019) of babies born <32 weeks and 26% (3525/13527) born <27 weeks received postnatal corticosteroids for BPD. Dexamethasone was predominantly used 5.3% (3309/62019), followed by late hydrocortisone 1.5%, inhaled budesonide 1.5%. prednisolone 0.8%, early hydrocortisone 0.3% and methylprednisolone 0.05%. Dexamethasone use increased over time (2012: 4.5 vs 2019: 5.8%, p=0.04). Median postnatal age of initiation of corticosteroid course was around 3 weeks for late hydrocortisone, 4 weeks for dexamethasone, 6 weeks for inhaled budesonide, 12 weeks for prednisolone and 16 weeks for methylprednisolone. Babies who received postnatal corticosteroids were born more prematurely, had a higher incidence of comorbidities and a longer length of stay.

**Conclusions** In England and Wales, around 1 in 12 babies born less than 32 weeks and 1 in 4 born less than 27 weeks receive postnatal corticosteroids to prevent or treat BPD. Given the lack of convincing evidence of efficacy, challenges of recruiting to and length of time taken to conduct randomised controlled trial, our data highlight the need to monitor long-term outcomes in children who received neonatal postnatal corticosteroids.

## STRENGTHS AND LIMITATIONS OF THIS STUDY

⇒ Data for 62 019 preterm babies were included in this study.

⇒ Exposure to different types of postnatal corticosteroids, including timing of initiation and duration are described.

⇒ Data over an 8-year period enabled time trends to be evaluated.

⇒ Lack of data on the indication for postnatal corticosteroid may lead to overestimation of babies exposed to hydrocortisone for bronchopulmonary dysplasia versus for blood pressure or adrenal insufficiency.

⇒ We were unable to report cumulative doses of dexamethasone as this was not available.

## WHAT IS KNOWN ABOUT THE SUBJECT

⇒ Bronchopulmonary dysplasia is the most common complication of preterm birth.

⇒ The use, choice and timing of postnatal corticosteroid remains controversial due to the increased risk of neurodevelopmental impairment associated with high-dose steroids.

## WHAT THIS STUDY ADDS

⇒ Between 2012 and 2019 in England and Wales, around 8% of babies born <32 weeks received postnatal corticosteroid; use was highest in the most preterm with one in four of babies born <27 gestational weeks given postnatal corticosteroid (PNC).

⇒ The most commonly used PNC was dexamethasone (5%).

⇒ Use of dexamethasone, hydrocortisone and inhaled budesonide has increased over time, while the use of prednisolone and methylprednisolone remained stable or declined.

## INTRODUCTION

Bronchopulmonary dysplasia (BPD), defined as needing respiratory support at 36 weeks postmenstrual age, is the most common adverse outcome of very preterm delivery, affecting up to 75% of babies born before 28 weeks of gestation worldwide.[1 2] In the most severely affected, supplemental oxygen at home is needed, pulmonary hypertension

may develop and lung-function abnormalities can persist into adulthood. BPD is also associated with delayed brain maturation and diffuse white matter anomalies that are associated with increased risk of neurodevelopmental impairment.[3] Despite increased use of less invasive ventilation after birth, there has been no significant decline in BPD rates globally, and nor improvement in lung function in childhood over time.[4] The pathogenesis of BPD remains to be fully understood but involves arrest of lung development (alveolar hypoplasia or dysmorphic pulmonary vasculature) and pulmonary inflammation. It is therefore postulated that the anti-inflammatory action of postnatal corticosteroids (PNC) may improve gas exchange and lung mechanics, thus facilitating weaning from invasive mechanical ventilation and improving respiratory outcomes.[5] However, the use of PNC remains controversial because of fear of long-term adverse neurodevelopmental effects, hypertension, intestinal perforation and impaired glucose metabolism.[6–8] While early use of dexamethasone (<7 days) is associated with increased incidence of cerebral palsy,[9] there is no strong evidence from available studies to suggest that later (>7 days) administration of lower dose dexamethasone, or early prophylactic use of low dose hydrocortisone, causes neurodevelopmental impairment.[10 11]

A recent systematic review and network meta-analysis included 62 studies involving 5559 neonates with a mean gestational age of 26 weeks and examined 14 different PNC regimens. This found that a moderately early-initiated (8–14 days), medium cumulative dose of systemic dexamethasone (2–4 mg/kg) was the best regimen for preventing mortality or BPD but acknowledged the weak evidence on which this recommendation is based.[12] Hence, when considering PNC treatment clinicians have to balance the adverse consequences of long-term ventilation with the potential for long-term neurodisability.

International studies report varying use of PNC (3%–50%) including both systemic and inhaled PNC for preterm neonates.[13 14] Policy recommendations regarding the use of dexamethasone or other corticosteroids, including hydrocortisone and inhaled corticosteroids, differ widely.[15–17] We aimed to describe the use of PNC for prevention or treatment of BPD over an 8-year period in very preterm babies born 2012–2019 in England and Wales, using the National Neonatal Research Database (NNRD). Objectives: (1) to quantify the number and proportion of babies who received PNC, timing and duration, (2) to describe the clinical characteristics of babies and unadjusted outcomes for babies who do and do not receive PNC

## METHODS

We undertook a retrospective and descriptive cohort study using deidentified, routinely recorded neonatal clinical data held in the NNRD. The NNRD is a national resource containing detailed, quality assured, clinical information extracted from the electronic patient records of admissions to National Health Service neonatal units.[18 19]

### Study population

We included data held in the NNRD on all babies born less than 32 weeks gestation born and admitted to neonatal care units in England and Wales between 1 January 2012 and 31 December 2019. We excluded babies with missing gestational ages. We extracted variables related to maternal factors (ethnicity, age, antenatal steroids); infant demographics (birth weight, sex, gestational age), clinical characteristics (condition at birth including Apgar score and resuscitation details); PNC use including methylprednisolone, prednisolone, hydrocortisone, dexamethasone, budesonide; respiratory support, comorbidities (gastrointestinal perforation, BPD, retinopathy of prematurity (ROP) that required treatment, brain injury on imaging)[20 21] and survival to discharge from neonatal care. Data were extracted by using SAS V.13.

### Definitions

PNC may be used for indications other than prevention or treatment of BPD (upper airway obstruction, blood pressure); however, indication for postnatal steroid use is not held in the NNRD. We, therefore, applied the following pragmatic definitions to identify babies we considered to have received postnatal steroids for BPD prevention or treatment:

Dexamethasone: Dexamethasone for ≥5 consecutive days (5 days was chosen to avoid the inclusion of babies given a shorter course for upper airway obstruction).

Early hydrocortisone: Hydrocortisone started on postnatal day 1 or 2 and given for >7 consecutive days consistent, with the PREMILOC trial.[22]

Late hydrocortisone: Hydrocortisone started after postnatal day 2 and given for >2 consecutive and >7 days total duration (to differentiate this from hydrocortisone given for hypotension).

Prednisolone: Prednisolone is given for ≥3 consecutive days.

Methylprednisolone: Methylprednisolone is given for ≥3 consecutive days.

Inhaled budesonide: Use of inhaled budesonide for ≥1 day.

BPD was defined as receiving any respiratory or ventilatory support or supplemental oxygen at 36 weeks postmenstrual age,[23] treatment for ROP was defined as cryotherapy, laser therapy or injection of antivascular endothelial growth factor therapy for ROP in either or both eyes, severe necrotising enterocolitis was defined as necrotising enterocolitis resulting in surgery or confirmed at surgery,[24] brain injury was defined as either left or right grade 3 or higher intraventricular haemorrhage or periventricular leukomalacia,[20] gastrointestinal perforation was defined as having a perforation if presence of relevant diagnostic and/or procedural codes (online supplemental file 1).

## Data analysis

Simple descriptive statistics were used to describe the study population, clinical characteristics, survival and comorbidities without adjustment. We report results for the entire cohort of babies born <32 weeks and also a subgroup of the most preterm babies born <27 weeks gestation based on the British Association of Perinatal Medicine extreme preterm criteria.[25] Data were described using medians and IQRs for continuous and non-parametric (as determined by a Shapiro-Wilk normality test) data and proportions for categorical data. Temporal trends were assessed using linear regression. For all the statistical results, a p<0.05 was considered as statistically significant. Analyses were performed using RStudio, SPSS V.25 and GraphPad Prism V.5.

## Patient and public involvement statement

Patients or the public were not involved in the design, or conduct, or reporting, or dissemination plans of our research.

## RESULTS

### Number and proportion of babies over time who received PNC for BPD

A total of 62019 babies born <32 weeks gestation were born and admitted to neonatal units in England and Wales during the study period. A total of 8140 (13%) babies were recorded as having received PNC for any indication and 7.6% (4713/62019) received at least one type of PNC for BPD (figure 1). The most commonly used PNC was dexamethasone 5.3% (3309/62019), followed by late hydrocortisone 1.5% (954/62019), inhaled budesonide 1.5% (922/62019), prednisolone 0.8% (500/62019). Early hydrocortisone 0.3% (180/62019) and methylprednisolone 0.05% (32/62019) use were rare. 6.3% (3914/62019) received only one type of PNC (early hydrocortisone 156, late hydrocortisone 620, dexamethasone 2461, prednisolone 196, inhaled budesonide 481). 1.3% (799/62019) received more than one type of PNC.

Between 2012 and 2019, an increasing proportion of very preterm babies received PNC (p=0.005) (figure 2). There was an overall increase use over time for dexamethasone, inhaled budesonide and late hydrocortisone even though there was a small decrease between 2018 and 2019; dexamethasone (2012: 4.5% vs 2018: 6.6% vs 2019: 5.8%, 2.0% increase per year, p=0.04); inhaled budesonide (2012: 0.6% vs 2018: 2.5% vs 2019: 2.4%, 19.3% increase per year, p<0.001) (922 babies); late hydrocortisone (2012: 1.2% vs 2018: 2.0% vs 2019: 1.7%, 4.1% increase per year, p=0.02). The number for early hydrocortisone was very small (180) and its use was first seen in 2017 and has increased over time (2012: 0.1% vs 2019: 0.8%, 3.7% increase per year, p=0.05). Use of

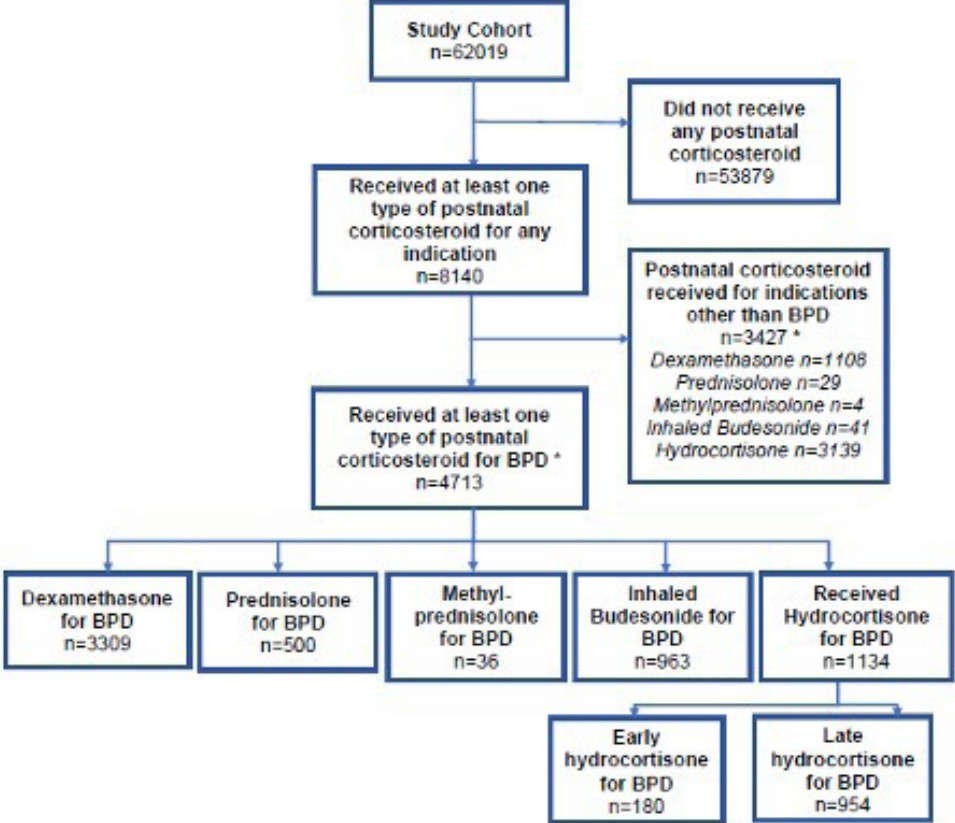

**Figure 1** Flow chart of cohort who received postnatal steroids. *Babies can have more than one postnatal corticosteroid type. BPD, bronchopulmonary dysplasia.

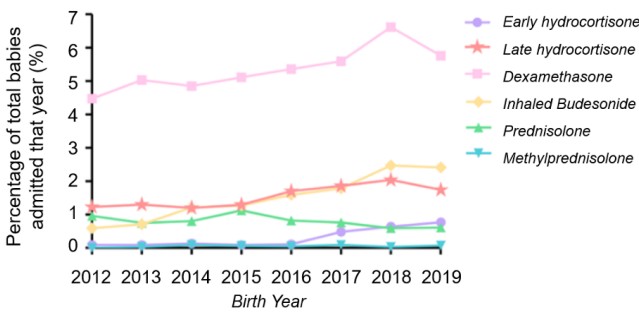

**Figure 2** Graph showing the use of postnatal corticosteroids over time.

prednisolone has decreased (2012: 1.0% vs 2019: 0.6%, 7.8% decrease per year, p<0.09) (500 babies) and methylprednisolone remained similar over time (2012: 0.0% vs 2019: 0.1%).

### Clinical characteristics of babies overall and by type of PNC for BPD

The descriptive and clinical characteristics of babies by type of PNC are presented in table 1. The median (IQR) gestational age for babies who received PNC was 25[24-27] weeks; 71% received a complete course of antenatal steroids; 87% were intubated at birth. Overall among the entire cohort less than 32 weeks, the intubation rate at delivery was 51% (31 609/62 019) and 92% for less than 27 weeks (12 480/13 527) (table 2).

At the time of PNC initiation, 75% and 20% were receiving invasive and non-invasive ventilatory support, respectively, with the exception of inhaled budesonide where 49% were on invasive ventilation; 7% received drugs for resuscitation or cardiac massage at delivery.

### Timing and duration of PNC for BPD

Median (IQR) postnatal age at initiation was 19 (8–40) days for late hydrocortisone, 28 (19–45) days for dexamethasone, 42 (30–70) days for inhaled budesonide, 865 (73–101) for prednisolone and 116 (90–155) days for methylprednisolone (table 1). Over time, the postnatal age at initiation for all PNC remained fairly consistent except for inhaled budesonide which is being used earlier in recent years (2012 median (IQR) 79 (57–116) vs 2019 30 (14–51)) (figure 3). The duration of use was shortest for methylprednisolone 7 (4–11) days and longest for late hydrocortisone 20 (11–38) days (table 1).

### Descriptive clinical characteristics, comorbidities and survival by gestational age groups

The clinical characteristics, comorbidities and survival in babies who did and did not receive PNC are presented by gestational subgroups (<32 weeks and <27 weeks) in table 2. A higher proportion of more preterm babies received PNC (<27 weeks 26% (3525/13 527) vs<32 weeks 8% (4713/58 592)). Of those who received PNC, the median (IQR) days to extubation after PNC administration were similar around 5 (2–13) among<32 and <27 gestational week groups (table 2). Babies who received

PNC had a higher incidence of comorbidities and a longer length of stay compared with those who did not (table 2).

For babies born <27 week, there was higher survival for babies who received versus did not receive PNC (87% vs 70%). However, 80% of deaths among those who did not receive PNC occurred within the first 2 weeks after birth (median (IQR) postnatal day of death 3 (1–12) days) compared with 5% of deaths occurring within first 2 weeks among those who received PNC (postnatal day of death 48 (29–96) days).

## DISCUSSION

We show that in England and Wales, PNC use in preterm babies to prevent or treat BPD, increased between 2012 and 2019 with about one in four babies born less than 27 weeks gestation receiving a steroid, most commonly dexamethasone, at a postnatal age of around 3 weeks. Late hydrocortisone and inhaled budesonide were the next most used, with early hydrocortisone administered very rarely. Our findings reflect current guidance from the UK National Institute for Health and Care Excellence, which recommends limiting the use of dexamethasone to babies after the age of 8 days to assist in weaning from ventilatory support, and not as 'prophylaxis' in younger infants. No recommendation is provided for hydrocortisone or nebulised budesonide, given limited evidence.[8]

The increase in proportion of infants exposed to PNC is not explained by an increase in extremely preterm infants admitted. The median gestational age in the less than 27 weeks group remained at 25.4 weeks between 2012 and 2019, and the proportion of infants born<27 weeks/<32 weeks (2012: 22% vs 2019: 21%) remained similar over time. The increase is also not explained by survivor bias, as the mortality rates before 28 days (the median postnatal age of administering the most commonly used PNC dexamethasone) were similar (2012: 7.8%) vs 2019: (6.8%).

The strengths of our study include the large, population representative population of all preterm babies born in England and Wales over an 8-year period. This allowed time trends to be evaluated and for findings to be widely applicable to UK neonatal units. The main limitation of this study is the lack of data on the indication for PNC. The pragmatic definition of dexamethasone ≥5 consecutive days is likely to capture dexamethasone use for BPD versus upper airway pathology. However, the definition for late hydrocortisone (started after postnatal day 2 and given for >2 consecutive and >7 days total duration), may still misclassify babies given hydrocortisone for blood pressure support or adrenal insufficiency and may therefore have resulted in an overestimation of the number of babies exposed to PNC for BPD treatment or prevention. The possible misclassification of PNC for BPD may also explain the unlikely finding that around 5% of infants were not on any respiratory support (including oxygen) on the day of initiation of PNC. Erroneous or missing daily data on the type of respiratory support could also

**Table 1** Clinical characteristics of infants by type of postnatal corticosteroid <32 weeks gestation

| | Dexamethasone (n=3309) | Late hydrocortisone (n=954) | Prednisolone (n=500) | Methyl-prednisolone (n=32) | Inhaled budesonide (n=922) | Early hydrocortisone (n=180) |
|---|---|---|---|---|---|---|
| Gestational age in weeks, median (IQR) | 25.1 (24.3–26.6) | 25.4 (24.3–27.6) | 25.4 (24.5–26.7) | 24.9 (24.1–28.0) | 25.7 (24.6–27.3) | 25.0 (24.0–27.0) |
| Antenatal corticosteroids | | | | | | |
| None | 166 (5%) | 43 (5%) | 26 (5%) | 2 (6%) | 25 (3%) | 3 (2%) |
| Complete | 2306 (70%) | 677 (71%) | 344 (69%) | 24 (75%) | 699 (76%) | 124 (67%) |
| Incomplete | 574 (17%) | 165 (17%) | 98 (20%) | 5 (16%) | 151 (16%) | 37 (32%) |
| Missing | 263 (8%) | 67 (7%) | 32 (6%) | 1 (3%) | 46 (5%) | – |
| Birth weight (g) median (IQR) | 710 (620–840) | 700 (600–854) | 710 (615–84) | 755 (597–869) | 758 (640–910) | 757 (645–916) |
| Sex | | | | | | |
| Male | 1975 (60%) | 591 (62%) | 288 (58%) | 22 (69%) | 580 (61%) | 93 (50%) |
| Apgar 5 mins | | | | | | |
| ≤ 5 | 839 (25%) | 231 (24%) | 115 (23%) | 9 (28%) | 218 (24%) | 61 (33%) |
| >5 | 1960 (59%) | 602 (63%) | 291 (58%) | 20 (63%) | 621 (67%) | 100 (54%) |
| Missing | 510 (15%) | 121 (13%) | 94 (19%) | 3 (9%) | 83 (9%) | 23 (13%) |
| Cardiac massage at delivery | 245 (7%) | 64 (7%) | 31 (6%) | 3 (9%) | 52 (6%) | 18 (10%) |
| Drugs for resuscitation at delivery | 255 (8%) | 72 (8%) | 50 (10%) | 2 (6%) | 50 (5%) | 20 (11%) |
| Intubation at delivery | 3021 (91%) | 838 (88%) | 448 (90%) | 28 (88%) | 698 (76%) | 162 (88%) |
| Postnatal age (days) at initiation of postnatal corticosteroid, median (IQR) | 28 (19–45) | 19 (8–40) | 86 (73–101) | 116 (90–155) | 42 (23–70) | 1 (1–2) |
| Duration (days), median (IQR) | 14 (10–25) | 20 (11–38) | 9 (8–30) | 7 (4–11) | 19 (9–34) | 11 (10–14) |
| Respiratory support at time of steroid initiation | 2799 (85%) | 718 (75%) | 372 (74%) | 26 (81%) | 448 (49%) | 171 (93%) |
| Invasive ventilation | 445 (13%) | 145 (18%) | 86 (17%) | 4 (13%) | 422 (46%) | 12 (7%) |
| Non-invasive support | 64 (2%) | 90 (9%) | 42 (8%) | 2 (6%) | 51 (6%) | 1 (0.5%) |
| None of the above | | | | | | |

**Table 2** Clinical characteristics, survival and comorbidities of babies who did and did not receive postnatal steroids presented by <32 and <27 weeks gestation groups

| | No postnatal steroids given for BPD, born <32 weeks GA | Postnatal steroids for BPD, born <32 weeks GA | No postnatal steroids given for BPD, born <27 weeks GA | Postnatal steroids for BPD, born <27 weeks |
|---|---|---|---|---|
| | n=62 019 | | n=13 527 | |
| n (%) | 57 306 (92%) | 4713 (8%) | 10 002 (74%) | 3525 (26%) |
| GA at birth (weeks) | 29 | 25.4 | 25.3 | 24.9 |
| Median (IQR) | (27.2–31.4) | (24.4–27.0) | (24.1–26.5) | (24.1–25.7) |
| Birth weight (g) | 1230 | 720 | 760 | 690 |
| Median (IQR) | (946–1510) | (620–860) | (648–880) | (607–795) |
| Male sex | 31 117 (54%) | 2786 (59%) | 5301 (53%) | 2030 (58%) |
| Intubation at birth | 27 509 (48%) | 4100 (87%) | 9202 (92.3%) | 3278 (93%) |
| Antenatal corticosteroids | | | | |
| None | 3256 (6%) | 220 (5%) | 764 (8%) | 170 (1%) |
| Complete | 39 997 (70%) | 3343 (71%) | 6754 (68%) | 2434 (69%) |
| Incomplete | 10 574 (18%) | 807 (17%) | 2235 (22%) | 659 (17%) |
| Missing | 3479 (6%) | 343 (7%) | 249 (2%) | 262 (7%) |
| Apgar 5 mins | | | | |
| ≤ 5 | 8300 (14%) | 1138 (24%) | 2699 (27%) | 919 (26%) |
| >5 | 41 445 (72%) | 2880 (61%) | 6420 (64%) | 2091 (59%) |
| Missing | 7561 (13%) | 695 (15%) | 900 (9%) | 515 (15%) |
| Cardiac massage at delivery | 2441 (4%) | 315 (7%) | 844 (8%) | 232 (2%) |
| Total length of stay (days) | 47 | 113 | 88 | 118 |
| Median (IQR) | (33–70) | –92 to 137 | –60 to108 | –99 to 141 |
| Total days mechanical ventilation (days) | 2 (0–5) | 33 (20–48) | 17 (4–24) | 37 (25–51) |
| Median (IQR) | | | | |
| Days to extubation after PNC administered Median (IQR) | NA | *5 (2–13) | NA | †5 (2–12) |
| Extubation | NA | | NA | |
| 1.Extubated and survived | | 1.2768 (59%) | | 1.2160 (61%) |
| 2.Extubated and died | | 2.546 (12%) | | 2.420 (12%) |
| 3.Not extubated after intubation* | | 3.1333 (28%) | | 3.927 (26%) |
| 4.Not intubated | | 4. 66 (1%) | | 4.18 (1%) |
| Survival to discharge | 52 498 (91.6%) | 4098 (87.0%) | 6981 (69.8%) | 3052 (86.6%) |
| BPD status | 15 473 (27%) | 4289 (91.0%) | 7101 (71.0%) | 3285 (93.2%) |
| Treated Retinopathy of prematurity | 1247 (2.2%) | 1302 (27.6%) | 880 (8.8%) | 1166 (33.0%) |
| Severe NEC | 1143 (2.0%) | 693 (14.7%) | 643 (6.4%) | 296 (8.4%) |
| GI perforation | 877 (1.5%) | 290 (6.2%) | 536 (5.4%) | 253 (7.2%) |
| Brain injury on imaging | 2212 (3.9%) | 443 (9.4%) | 1105 (11.1%) | 377 (10.7%) |

*A total of 3314 babies were extubated following PNC.
†A total of 2580 babies <27 weeks were extubated following PNC
BPD, bronchopulmonary dysplasia; GA, gestational age ; GI, gastrointestinal; NEC, necrotising enterocolitis; PNC, postnatal corticosteroid.

explain this finding. As the NNRD contains routine clinical data, we also cannot exclude the possibility of missing data leading to an underestimation of PNC use. We were unable to report cumulative doses of dexamethasone, as this is not available in the NNRD.

Studies in other countries have reported similar use of PNC. The 2011 EPICE study found that 13.9% of babies born 24–29 weeks received PNC for BPD, but rates varied from 3.1% (Portugal) to 49.4% (Germany).[26] In a randomised controlled trial (RCT) of erythropoietin versus placebo involving 30 units in the USA, 38% of babies born 24 to 27+6 weeks who survived to discharge were treated with PNC.[27] The median (IQR) postnatal age for dexamethasone initiation was very similar to our study 29 (20–44) days with a similar duration 10 (5–15) days, but use of prednisolone or methylprednisolone was

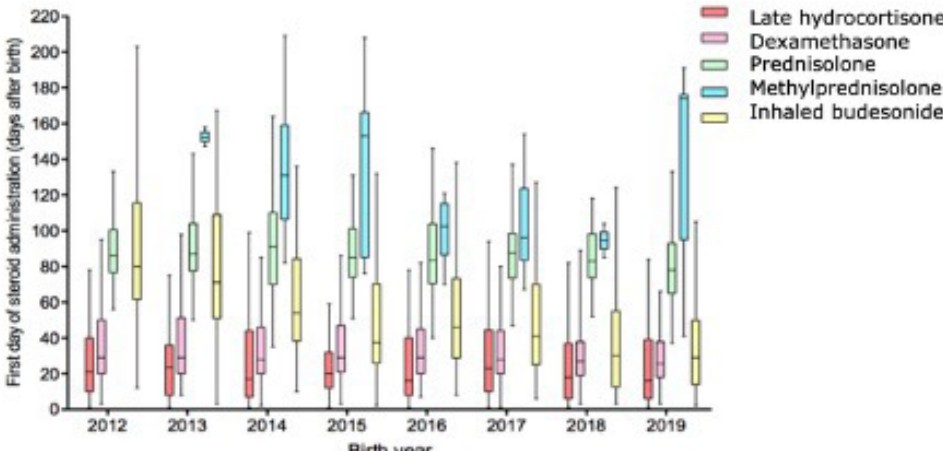

**Figure 3** A boxplot showing the median, IQR and range of first day postnatal corticosteroid administration for BPD (grouped by postnatal corticosteroid type) in babies who were born at <32 weeks gestation between 2012 and 2019 and were admitted to neonatal units in England and Wales. BPD, bronchopulmonary dysplasia.

earlier. Studies in other countries have also reported the use of inhaled corticosteroids for BPD (25% and 80% in the USA[28] and Japan,[29] respectively).

The overall incidence of BPD in the entire study cohort <32 weeks was 31%, and 91% and 27% for babies who did and did not receive PNC, respectively. Given the observational nature of this study, it is unsurprising that babies who received PNC were more preterm and sicker. We did not apply causal inference methods such as propensity matching to examine the associations between PNC and outcomes because achieving good matching was unlikely given the marked differences in characteristics between babies who did and did not receive PNC. Survival was higher among babies born <27 weeks who received PNC, likely due to survival bias. This occurs when patients are expected to survive until the exposure. Patients who die before they have had the opportunity to be exposed will be 'unexposed' by definition, and this introduces an artificial survival advantage associated with the exposed subjects regardless of treatment effectiveness.[30] The majority of deaths occurred in the first 2 weeks following birth, prior to the median age of dexamethasone initiation (19 days).

PNCs are one of the most investigated medications in neonatology, with over 80 RCT enrolling over 9000 babies. Despite multiple trials and systematic reviews,[10 11 31 32] there remains uncertainty about the appropriate formulation, timing, dosage and duration of PNC. Further RCT on dosing and timing of dexamethasone are unlikely to be feasible given that pilot studies, such as the MINIDEX study did not demonstrate feasibility,[33] and the DART study[34 35] failed to recruit target numbers and was abandoned. The recently published results of an RCT involving 800 babies born less than 30 weeks showed that a 10-day course of hydrocortisone at 14–28 days in babies who have been intubated for at least 7 days, did not result in substantially higher survival without moderate or severe BPD compared with placebo. They also showed that survival without moderate or severe neurodevelopmental impairment did not differ substantially between the two groups.[36] Given the lack of benefit, and potential harms such as gastrointestinal perforation, courses of hydrocortisone at 2–3 weeks old to treat BPD should probably be avoided.

We found increasing use of inhaled budesonide in our study at around 6 weeks of age. A major rationale for inhaled corticosteroids is to avoid some of the complications of systemic corticosteroids by concentrating the drug in the lung. However, given the lack of evidence for early or later inhaled corticosteroids, they are not recommended for routine clinical practice.[37 38] Of note, despite widespread concern regarding long-term effects, few studies have been powered to examine neurodevelopmental and cognitive outcomes. Although less meaningful to parents, and a poor predictor of future functional impairment, most PNC studies have used BPD (oxygen dependency at 36 weeks corrected) as a primary outcome.

We have shown that around 400 babies each year are exposed to postnatal dexamethasone and the majority survive to discharge. We suggest that long-term neurodevelopmental surveillance should be considered mandatory in these infants, and an essential component of good clinical practice. Linking together routine health and educational datasets could be a potential, more affordable and practical solution.

## CONCLUSION

In England and Wales, around 1 in 12 babies born less than 32 weeks and 1 in 4 born less than 27 weeks receive PNCs to prevent or treat BPD. Given the lack of convincing evidence of efficacy, challenges of recruiting to and length of time taken to conduct RCT, our data highlight the need to monitor long-term outcomes in children who receive PNCs.

**Acknowledgements** We wish to acknowledge all neonatal units who contribute data to the NNRD, known collectively as the UK Neonatal collaborative www.imperial.ac.uk/neonatal-data-analysis-unit/neonatal-data-analysis-unit/list-of-national-neonatal-units/.

**Collaborators** We wish to acknowledge the UK Neonatal Collaborative comprising neonatal units contributing data to the National Neonatal Research Database (NNRD) www.imperial.ac.uk/neonatal-data-analysis-unit/neonatal-data-analysis-unit/contributing-to-the-national-neonatal-research-database/. UK Neonatal Collaborative (UKNC): Dr Matthew Babirecki, (Airedale General Hospital), Dr Anand Kamalanathan, (Arrowe Park Hospital), Dr Clare Cane, (Barnet Hospital), Dr Kavi Aucharaz, (Barnsley District General Hospital), Dr Aashish Gupta, (Basildon Hospital), Dr Alistair Ewing, (Basingstoke & North Hampshire Hospital), Dr L M Wong, (Bassetlaw District General Hospital), Dr Anita Mittal, (Bedford Hospital), Dr Lindsay Halpern, (Birmingham City Hospital), Dr Pinki Surana, (Birmingham Heartlands Hospital), Dr Matt Nash, (Birmingham Women's Hospital), Dr Sam Wallis, (Bradford Royal Infirmary), Dr Ahmed Hassan, (Broomfield Hospital, Chelmsford), Dr Karin Schwarz, (Calderdale Royal Hospital), Dr Shu-Ling Chuang, (Chelsea & Westminster Hospital), Dr Penelope Young, (Chesterfield & North Derbyshire Royal Hospital), Dr Ramona Onita, (Colchester General Hospital), Dr Graham Whincup, (Conquest Hospital), Dr Joanne Dangerfield, (Countess of Chester Hospital), Dr Jocelyn Morris, (Croydon University Hospital), Dr Yee Aung, (Cumberland Infirmary), Dr Abdul Hasib, (Darent Valley Hospital), Dr Mehdi Garbash, (Darlington Memorial Hospital), Dr Alex Allwood, (Derriford Hospital), Dr Pauline Adiotomre, (Diana Princess of Wales Hospital), Dr Nigel Brooke, (Doncaster Royal Infirmary), Dr Abby Deketelaere, (Dorset County Hospital), Dr Toria Klutse, (East Surrey Hospital), Dr Sonia Spathis, (Epsom General Hospital), Sathish Krishnan, (Frimley Park Hospital), Dr Samar Sen, (Furness General Hospital), Dr Jez Jones, (George Eliot Hospital), Dr Geedi Farah, (Glan Clwyd Hospital), Dr Prem Pitchaikani, (Glangwili General Hospital), Dr Jennifer Holman, (Gloucester Royal Hospital), Dr Pinki Surana, (Good Hope Hospital), Dr Stanley Zengeya, (Great Western Hospital), Dr Geraint Lee, (Guy's & St Thomas' Hospital), Dr Sobia Balal, (Harrogate District Hospital), Dr Cath Seagrave, (Hereford County Hospital), Dr Tristan Bate, (Hillingdon Hospital), Dr Hilary Dixon, (Hinchingbrooke Hospital), Dr Narendra Aladangady, (Homerton Hospital), Dr Hassan Gaili, (Hull Royal infirmary), Dr Matthew James, (Ipswich Hospital), Dr M Lal, (James Cook University Hospital), Dr Oluseun Tayo, (James Paget Hospital), Dr Poornima Pandey, (Kettering General Hospital), Dr Ravindra Bhat, (Kings College Hospital), Dr Simon Rhodes, (King's Mill Hospital), Dr Jonathan Filkin, (Kingston Hospital), Dr Savi Sivashankar, (Lancashire Women and Newborn Centre), Dr Lawrence Miall, (Leeds Neonatal Service), Dr Jonathan Cusack, (Leicester General Hospital), Dr Venkatesh Kairamkonda, (Leicester Royal Infirmary), Dr Michael Grosdenier, (Leighton Hospital), Dr Ajay Reddy, (Lincoln County Hospital), Dr J Kefas, (Lister Hospital), Dr Alison Bedford Russell, (Liverpool Women's Hospital), Dr Jennifer Birch, (Luton & Dunstable Hospital), Dr Gail Whitehead, (Macclesfield District General Hospital), Dr Ashok Karupaiah, (Manor Hospital), Dr Ghada Ramadan, (Medway Maritime Hospital), Dr I Misra, (Milton Keynes General Hospital), Dr Nicola Johnson, (Musgrove Park Hospital), Dr Richard Heaver, (New Cross Hospital), Dr Mohammad Alam, (Newham General Hospital), Dr Prakash Thiagarajan, (Nobles Hospital), Dr Priya Muthukumar, (Norfolk & Norwich University Hospital), Dr Tiziana Fragapane, (North Devon District Hospital), Dr Ngozi Edi-Osagie, (North Manchester General Hospital), Dr Cheentan Singh, (North Middlesex University Hospital), Dr Subodh Gupta, (Northampton General Hospital), Jess Reynolds, (Northumbria Specialist Emergency Care Hospital), Dr Khadija Ben-Sasi, (Northwick Park Hospital), Dr Steven Wardle, (Nottingham City Hospital), Dr Steven Wardle, (Nottingham University Hospital (QMC)), Dr Victoria Nesbitt, (Ormskirk District General Hospital), Dr Eleri Adams, (Oxford University Hospitals, John Radcliffe Hospital), Dr Katharine McDevitt, (Peterborough City Hospital), Dr Ruchika Gupta, (Pilgrim Hospital), Dr David Gibson, (Pinderfields General Hospital (Pontefract General Infirmary)), Prof Minesh Khashu, (Poole General Hospital), Dr Iyad Al-Muzaffar, (Prince Charles Hospital), Dr Kate Creese, (Princess of Wales Hospital), Dr Chinnappa Reddy, (Princess Alexandra Hospital), Dr Mark Johnson, (Princess Anne Hospital), Dr Prashanth Bhat, (Princess Royal Hospital), Dr Patricia Cowley, (Princess Royal Hospital (previously Royal Shrewsbury Hospital)), Dr Rashmi Gandhi, (Princess Royal University Hospital), Dr Charlotte Groves, (Queen Alexandra Hospital), Dr Lidia Tyszczuk, (Queen Charlotte's Hospital), Dr Shilpa Ramesh, (Queen Elizabeth Hospital, Gateshead), Dr Glynis Rewitzky, (Queen Elizabeth Hospital, King's Lynn), Mrs Julia Croft, (Queen Elizabeth Hospital, Woolwich - see notes), Dr Bushra Abdul-Malik, (Queen Elizabeth the Queen Mother Hospital), Dr Dominic Muogbo, (Queen's Hospital, Burton on Trent), Dr Ambalika Das, (Queen's Hospital, Romford), Dr Angela D'Amore, (Rosie Maternity Hospital, Addenbrookes), Dr Soma Sengupta, (Rotherham District General Hospital), Dr Christos Zipitis, (Royal Albert Edward Infirmary), Dr Peter De Halpert, (Royal Berkshire Hospital), Dr Archana Mishra, (Royal Bolton Hospital), Dr Chris Warren, (Royal Cornwall Hospital), Dr John McIntyre, (Royal Derby Hospital), Dr Nagendra Venkata, (Royal Devon & Exeter Hospital), Dr Lucinda Winckworth, (Royal Hampshire County Hospital), Dr Joanne Fedee, (Royal Lancaster Infirmary), Dr Anitha Vayalakkad, (Royal Oldham Hospital), Dr Raju Narasimhan, (Royal Preston Hospital), Dr Lee Abbott, (Royal Stoke University Hospital), Dr Ben Obi, (Royal Surrey County Hospital), Dr Prashanth Bhat, (Royal Sussex County Hospital), Dr Stephen Jones, (Royal United Hospital), Dr Richard Hearn, (Royal Victoria Infirmary), Dr Anjali Petkar, (Russells Hall Hospital), Dr Jim Baird, (Salisbury District Hospital), Dr Kirsten Mack, (Scarborough General Hospital), Dr Pauline Adiotomre, (Scunthorpe General Hospital), Dr Arun Ramachandran, (Singleton Hospital), Dr Vineet Gupta, (Southend Hospital), Dr Faith Emery, (Southmead Hospital), Dr Charlotte Huddy, (St George's Hospital), Dr Salim Yasin, (St Helier Hospital), Dr Akinsola Ogundiya, (St Mary's Hospital, IOW), Dr Lidia Tyszczuk, (St Mary's Hospital, London), Dr Ngozi Edi-Osagie, (St Mary's Hospital, Manchester), Dr Pamela Cairns, (St Michael's Hospital), Dr Vennila Ponnusamy, (St Peter's Hospital), Dr Victoria Sharp, (St Richard's Hospital), Dr Carrie Heal, (Stepping Hill Hospital), Dr Sanjay Salgia, (Stoke Mandeville Hospital), Dr Imran Ahmed, (Sunderland Royal Hospital), Dr Jacqeline Birch, (Tameside General Hospital), Dr Sunil Reddy, (The Grange University Hospital), Dr Porus Bastani, (The Jessop Wing, Sheffield), Dr Marice Theron, (The Royal Free Hospital), Dr Divyen Shah, (The Royal London Hospital - Constance Green), Dr Siba Paul, (Torbay Hospital), Dr Se-Yeon Park, (Tunbridge Wells Hospital), Dr Giles Kendall, (University College Hospital), Dr Puneet Nath, (University Hospital Coventry), Mrs Julia Croft, (University Hospital Lewisham), Dr Mehdi Garbash, (University Hospital of North Durham), Dr Hari Kumar, (University Hospital of North Tees), Dr Nitin Goel, (University Hospital of Wales), Dr Chris Rawlingson, (Victoria Hospital, Blackpool), Dr Delyth Webb, (Warrington Hospital), Dr Sue Bird, (Warwick Hospital), Dr Sankara Narayanan, (Watford General Hospital), Dr Yee Aung, (West Cumberland Hospital), Dr Elizabeth Eyre, (West Middlesex University Hospital), Dr Jageer Mohammed, (West Suffolk Hospital), Dr Sanjay Jaisal, (Wexham Park Hospital), Dr Caroline Sullivan, (Whipps Cross University Hospital), Dr Ros Garr, (Whiston Hospital), Dr Wynne Leith, (Whittington Hospital), Dr Vimal Vasu, (William Harvey Hospital), Dr Anna Gregory, (Worcestershire Royal Hospital), Dr Katia Vamvakiti, (Worthing Hospital), Dr Brendan Harrington, (Wrexham Maelor Hospital), Dr Ngozi Edi-Osagie, (Wythenshawe Hospital), Dr Megan Eaton, (Yeovil District Hospital), Dr Sundeep Sandhu, (York District Hospital), Dr Michael Cronin, (Ysbyty Gwynedd).

**Contributors** CB, CG and SU conceived this project. SY carried out the analyses. The first draft of the manuscript was written by SY and revised by CB, CG, SU and NM. All authors have reviewed and approved the final manuscript. CB is the guarantor and accepts full responsibility for the work and conduct of the study and had access to the data. All authors controlled the decision to publish.

**Funding** The funding for creating and maintaining the NNRD is from unrestricted funding awarded to NM. This includes costs involved in data transfer, storage, cleaning, merging, administration and regulatory approvals. SY undertook this project as part of the Imperial College Medical School Bachelor of Sciences programme. CB is funded by the United National Institute for Health Research through an Advanced Fellowship (NIHR 300617). CG is funded by the Medical Research Council (MRC) award.

**Competing interests** CG is funded by the UK Medical Research Council (MRC) through a Transition Support Award. He has received support from Chiesi Pharmaceuticals to attend an educational conference; in the past 5 years, he has been investigator on received research grants from Medical Research Council, National Institute of Health Research, Canadian Institute of Health Research, Department of Health in England, Mason Medical Research Foundation, Westminster Medical School Research Trust and Chiesi Pharmaceuticals. CG chairs the National Institute for Health and Care Research (NIHR) Research for Patient Benefit London Regional Panel. CB is funded by the UK NIHR through an Advanced Fellowship Award. She is deputy chair for the NIHR Prioritisation committee for Hospital-based care. CB has received support from Chiesi Pharmaceuticals to attend educational conferences.

**Patient and public involvement** Patients and/or the public were not involved in the design, or conduct, or reporting, or dissemination plans of this research.

**Patient consent for publication** Not applicable.

**Ethics approval** The study is covered by Research Ethics Committee Approval reference 16/LO/1093.

**Provenance and peer review** Not commissioned; externally peer reviewed.

**Data availability statement** Data are available on reasonable request. The data that support the findings of this study are available from the corresponding author CB, on reasonable request.

**Open access** This is an open access article distributed in accordance with the Creative Commons Attribution 4.0 Unported (CC BY 4.0) license, which permits others to copy, redistribute, remix, transform and build upon this work for any purpose, provided the original work is properly cited, a link to the licence is given, and indication of whether changes were made. See: https://creativecommons.org/licenses/by/4.0/.

**ORCID iDs**
Sabita Uthaya http://orcid.org/0000-0002-6112-2277
Chris Gale http://orcid.org/0000-0003-0707-876X
Neena Modi http://orcid.org/0000-0002-2093-0681
Cheryl Battersby http://orcid.org/0000-0002-2898-553X

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
