## [Reviewer comments · BMJ Open]

ARTICLE DETAILS

TITLE (PROVISIONAL)	Postnatal corticosteroid use for prevention or treatment of Bronchopulmonary Dysplasia in England and Wales 2012-2019: a retrospective population cohort study
AUTHORS	Yao, Sijia; Uthaya, Sabita; Gale, Chris; Modi, Neena; Battersby, Cheryl

VERSION 1 – REVIEW

REVIEWER	Nuytten, Alexandra Department of Neonatology, Jeanne de Flandre Hospital, Lille CHRU, Lille France
REVIEW RETURNED	23-May-2022

GENERAL COMMENTS	This is an interesting article for England and Wales, in order to update the epidemiology of PNS use. p6 line 33: PNC use rose between 2012 and 2019, I think the author should check if the GA changed too (are the infants more immature? That could explain the larger use?), and the mortality (was mortality higher in 2012 than 2019? Then the increase of dexamethasone use could be due to a survival bias?) p6 l59: 87% of infants were intubated at birth; it seems a lot to me, especially for not extremely preterm infants. I think the authors should comment this results p7 line 3: 75% of infants receiving PNC were under invasive ventilation, 20% under non invasive ventilation, and 5% not under any ventilatory support. This is unexpected, as it is recommended to use PNS to wean infants from ventilatory support. I think the author should comment this result. It would be interesting to have overall BPD status described. The EPICE cohort showed that BPD status was higher in UK than in other european regions (Nuytten et al, Neonatology 2020). The description of PNC use needs to be described with BPD in the population, so that extrapolation to another population can be discussed.
---

REVIEWER	Mohamed, Adel
REVIEW RETURNED	06-Jul-2022

GENERAL COMMENTS	Thank you very much for the opportunity to review this manuscript that I read with great interest. In this retrospective cohort study, investigators used the national database to describe and compare preterm infants (GA<32weeks' gestation) who do and do not receive
---

	postnatal corticosteroids for prevention or treatment of BPD. They found in England and Wales that around 1 in 12 babies born less than 32 weeks and 1 in 4 born less than 27 weeks receive postnatal corticosteroids to prevent or treat bronchopulmonary dysplasia. In addition, babies who received PNC were more preterm and sicker.
--	---

VERSION 1 – AUTHOR RESPONSE

1) Reviewer: 1

This is an interesting article for England and Wales, in order to update the epidemiology of PNS use.

p6 line 33: PNC use rose between 2012 and 2019, I think the author should check if the GA changed too (are the infants more immature? That could explain the larger use?), and the mortality (was mortality higher in 2012 than 2019? Then the increase of dexamethasone use could be due to a survival bias?)

Reply:

Thank you for your suggestion to include additional population descriptive characteristics over time, and agree that gestational age is an important influencing factor, which is the reason for presenting the descriptive characteristics (Table 2) in subgroups <27 and <32 weeks. We have added the following to the discussion to address the suggestion of changing population and survivor bias.

“The increase in proportion of infants exposed to PNC is not explained by an increase in extremely preterm infants admitted. The median gestational age in the less than 27 weeks group remained at 25.4 weeks between 2012 and 2019, and the proportion of infants born <27 weeks/<32 weeks (2012: 22% vs 2019: 21%) remained similar over time. The increase is also not explained by survivor bias as the mortality rates before 28 days (the median postnatal age of administering the most commonly used PNC dexamethasone) were similar (2012: 7.8%) vs 2019: (6.8%).”

2) p6 l59: 87% of infants were intubated at birth; it seems a lot to me, especially for not extremely preterm infants. I think the authors should comment this results

Reply: We wish to clarify that the infants who received PNC are mainly extremely preterm infants (median gestational age is 25) and this is included in the same sentence: *“The median (IQR) gestational age for babies who received PNC was 25 (24-27) weeks; 71% received a complete course of antenatal steroids; 87% were intubated at birth.”* Table 2 shows that 92-93% of all babies born less than 27 weeks (whether they receive postnatal steroids or not) are intubated at birth.

To improve clarity, we have done the following

1) Included additional data in Table 2 (intubation at birth, Agpar score at 5 mins, cardiac massage at delivery) for those who did not receive PNC for BPD in each subgroup (<32 weeks and <27 weeks).

2) added “by gestational age groups” to the subheading and the following sentence in the beginning of the paragraph “The clinical characteristics, co-morbidities and survival in babies who did and did not receive PNC are presented by gestational subgroups (<32 weeks and <27 weeks) in Table 2.”

3) p7 line 3: 75% of infants receiving PNC were under invasive ventilation, 20% under non-invasive ventilation, and 5% not under any ventilatory support. This is unexpected, as it is recommended to use PNS to wean infants from ventilatory support. I think the author should comment this result.

Response: Thank you for your suggestion. We agree this is unlikely and have added this to the discussion about the possible misclassification of indication for postnatal steroids.

“The possible misclassification of PNC for BPD may also explain the unlikely finding that around 5% of infants were not on any respiratory support (including oxygen) on the day of initiation of PNC. Erroneous or missing daily data on the type of respiratory support could also explain this finding”

4) It would be interesting to have overall BPD status described. The EPICE cohort showed that BPD status was higher in UK than in other European regions (Nuytten et al, Neonatology 2020). The description of PNC use needs to be described with BPD in the population, so that extrapolation to another population can be discussed.

Response:

The EPICE cohort were not directly comparable as the babies were born 24 to 29+6 weeks and included only three regions in England versus these data presented from the National Neonatal Research Database (NNRD) which represent all 13 regions in England and Wales.

We have added to the discussion: *“The overall incidence of BPD in the entire study cohort <32 weeks was 31%, and 91% and 27% for babies who did and did not receive PNC, respectively.”*

Reviewer: 2

Comments to the Author:

Thank you very much for the opportunity to review this manuscript that I read with great interest. In

this retrospective cohort study, investigators used the national database to describe and compare preterm infants (GA<32weeks' gestation) who do and do not receive postnatal corticosteroids for prevention or treatment of BPD. They found in England and Wales that around 1 in 12 babies born less than 32 weeks and 1 in 4 born less than 27 weeks receive postnatal corticosteroids to prevent or treat bronchopulmonary dysplasia. In addition, babies who received PNC were more preterm and sicker.

Response: Thank you for your comments